# Developing a new model for patient recruitment in mental health services: a cohort study using Electronic Health Records

Felicity Callard,[1] Matthew Broadbent,[2] Mike Denis,[3] Matthew Hotopf,[4] Murat Soncul,[5] Til Wykes,[6] Simon Lovestone,[7] Robert Stewart[8]

▶ Prepublication history and additional material is available. To view please visit the journal (http://dx.doi.org/10.1136/bmjopen-2014-005654).

For numbered affiliations see end of article.

**Correspondence to**
Dr Felicity Callard;
felicity.callard@durham.ac.uk

## ABSTRACT

**Objectives:** To develop a new model for patient recruitment that harnessed the full potential of Electronic Health Records (EHRs). Gaining access to potential participants' health records to assess their eligibility for studies and allow an approach about participation ('consent for contact') is ethically, legally and technically challenging, given that medical data are usually restricted to the patient's clinical team. The research objective was to design a model for identification and recruitment to overcome some of these challenges as well as reduce the burdensome (and/or time consuming) gatekeeper role of clinicians in determining who is appropriate or not to participate in clinical research.

**Setting:** Large secondary mental health services context, UK.

**Participants:** 2106 patients approached for 'consent for contact'. All patients in different services within the mental health trust are gradually and systematically being approached by a member of the clinical care team using the 'consent for contact' model. There are no exclusion criteria.

**Primary and secondary outcome measures:** Provision of 'consent for contact'.

**Results:** A new model (the South London and Maudsley NHS Trust Consent for Contact model (SLaM C4C)) for gaining patients' consent to contact them about research possibilities, which is built around a de-identified EHR database. The model allows researchers to contact potential participants directly. Of 2106 patients approached by 25 October 2013, nearly 3 of every 4 gave consent for contact (1560 patients; 74.1%).

**Conclusions:** The SLaM C4C model offers an effective way of expediting recruitment into health research through using EHRs. It reduces the gatekeeper function of clinicians; gives patients greater autonomy in decisions to participate in research; and accelerates the development of a culture of active research participation. More research is needed to assess how many of those giving consent for contact subsequently consent to participate in particular research studies.

## Strengths and limitations of this study

- A new model for gaining patients' consent to contact them about research possibilities, which is built around a robustly de-identified Electronic Health Records (EHR) database.
- Currently, three of every four patients approached are giving consent to contact using the model.
- The model gives patients greater autonomy in decisions about research participation.
- The model was developed through extensive patient/service user involvement and consultation.
- The study does not examine whether patients who have given consent to contact go on to consent to participation in particular research studies.

## INTRODUCTION
### Recruiting patients into health research

One of the most pressing issues for researchers needing to recruit patients is how best to identify who might be suitable for their research, at the same time as adhering to legal and ethical frameworks regarding confidentiality and privacy. Medical data are, in multiple jurisdictions, generally accessible to researchers only with prior patient consent, or if the data are adequately de-identified; this means, in short, that a researcher needs to gain consent from an individual *before* accessing her personal details to adjudicate suitability for the research and before contacting her to ask about potential participation. Access to individuals' medical records is usually restricted to members of the clinical team, who thereby act as intermediaries through which contact between researchers and potential research participants is commonly made. Such a process is, however, often cumbersome and time-consuming: busy clinical teams often do not have the time to assist with recruitment; the alternative path of

writing to each patient (eg, within a general practice) to ask if she would agree to researchers accessing her medical records to assess suitability for a study is no less onerous. Furthermore, should the clinician still be a gate-holder for the patient? Is this not a remnant of medical paternalism, outdated in today's healthcare system and research context with citizen science and direct-to-consumer research opportunities such as the uBiome[1] project? Clinicians' desire to protect patients can, through 'gatekeeping', end up effectively denying them the opportunity to participate in research.[2] Such a problem is particularly pertinent within a mental health context. In short, it is time to allow patients greater autonomy in decisions to take part in research by reducing the gatekeeper role of clinicians—not least because such gatekeeper functions can exacerbate inequities in who is able to participate in health research.

Amidst wide concern about difficulties in recruiting to clinical studies,[3] there are calls to transform the culture of patient participation in health research (eg, US National Institutes of Health (NIH) Road Map,[4] and England's National Health Service (NHS) Constitution).[5] The influential 2008 Data Sharing Review in the UK recommended that 'The NHS should develop a system to allow approved researchers to work with healthcare providers to identify potential patients, who may then be approached to take part in clinical studies for which consent is needed'.[6] There appears, however, to be a reluctance to mandate legislatively a move away from the current model in which researchers require consent before accessing medical records.

Meanwhile, quantitative and qualitative studies in a number of countries have indicated that significant proportions of the public are cautious about giving researchers access to their medical records for recruitment purposes without prior consent, even as the public remains supportive of health research.[7–9] A mass public engagement exercise (6000 people) in England, for example, found that only 34% of adults and 10% of young people agreed with extending the range of people with direct access to patient records to assist in recruitment to clinical trials.[10] Given this and the current legislative environment, it is likely that in the UK, at least, the current principles surrounding consent (ie, that researchers are not able to access a patient's records without consent unless they are a member of the patient's clinical team) will continue to underpin research governance for some time. There is therefore a need to find ways of negotiating this governance environment that would allow more efficient identification of, and better engagement with, potential research participants, and would empower patients or service users to make their own decisions regarding which, if any, research projects to learn more about.[11–13]

## Consent for contact

One response to this challenge has been the formulation of what has, in the last decade, come to be termed 'consent for contact' or 'consent to consent'. This, as the Academy of Medical Sciences puts it, comprises a 'mechanism … for individuals to give *generic consent to be contacted about suitable research opportunities*, before considering whether they consent to take part in a specific study' [italics added].[14] Notably, 'consent for contact' processes are being put in place in a range of health research facilities. A project in British Columbia, Canada has, since 2007, set up 'Permission to Contact' platforms in different outpatient health clinics (cancer, cardiac and maternal health), which have proved effective in enhancing enrolment into translational research projects.[15 16] In the UK, the UK Biobank project requires explicit consent in order both to access the medical records of those joining the project, as well as potentially to recontact these participants in the future.[17] While such generic consent is appealing for researchers, its operationalisation poses a number of ethical, sociological, governance-related and technical questions (box 1). There is growing interest in how EHRs might be used in this regard. To date, the use of EHRs for recruitment has relied on mechanisms through which a member of the clinical team of the patient identified—via pseudonymised searching—as potentially eligible for the study is alerted and invited to contact that patient. Such use therefore currently maintains the distinction between those designing the studies and those recruiting into the studies.[18–22]

That many of the questions surrounding the use of EHRs for research remain unresolved, at a conceptual and an empirical level, is demonstrated by the number of medical, bioethical and governance-oriented bodies currently reflecting on them.[14 23–25] Our paper takes these discussions forward through focusing on an

> **Box 1** 'Consent for contact': sociological, ethical and technical issues
>
> ► How does 'consent for contact' reshape relationships between treating clinicians, patients and researchers—in that the traditional role of clinician (as patient advocate and/or paternalistic patient protector) in relation to the researcher is downgraded in the emergence of a new kind of compact between patient and researcher?
> ► How does patients' giving of generic consent to be contacted affect how they subsequently respond to invitations to participate in specific research projects (ie, do they feel more of an onus to give consent here, too, having given consent once already)?
> ► Does 'consent for contact' encourage an assumption that willingness to participate in health research is a moral obligation —and if so, what are the ethical, clinical and sociological implications?
> ► How can Electronic Health Records be best used in developing consent for contact procedures? How should their use navigate complex questions regarding control, ownership and use of such data in relation to consent, authorisation and safe-keeping?

empirical advance: the model of consent for contact developed by the South London and Maudsley (SLaM) NHS Foundation Trust (hereafter: SLaM C4C). For whatever one's position regarding the merits or demerits of 'consent for contact', it is commonly agreed that 'further work is needed to provide guidance and models to enable appropriate access and identification of patients for research'.[26] SLaM C4C was endorsed by the UK's Information Governance Review in 2013 as an exemplar of 'an approach that allows appropriate individuals to be identified and approached to take part, without giving researchers direct access to identifiable information before consent is obtained'.[24]

In the paper we present: (1) the specifics of SLaM C4C, which we believe to be the first successful implementation of consent for contact which both harnesses the potential of de-identified Electronic Health Records (EHRs) to expedite recruitment to research, and allows researchers to contact potential participants directly; (2) descriptive statistics on how SLaM C4C is being received by patients.

## METHODS
### The SLaM C4C model
SLaM C4C is designed to enhance patients' access to opportunities to participate in research projects of interest to them (England's NHS Constitution pledges 'to inform [patients] of research studies in which [they] may be eligible to participate').[5] It also enables researchers to identify and approach potentially eligible people. (These researchers have undergone rigorous approval procedures, which include being bound by the duties associated with a contract with King's Health Partners [an Academic Health Sciences Centre], which in turn involves various Human Resources checks, including a criminal record check in accordance with national Department of Health standards). The model comprises technical and procedural elements built into the EHR case register at the NIHR Specialist Biomedical Research Centre for Mental Health at the South London & Maudsley (SLaM) NHS Foundation Trust (hereafter: SLaM BRC). It was designed for use across SLaM, which is one of the largest mental healthcare providers in Europe, serving a local population of 1.2 million people, and including inpatient wards, outpatient and community services. Prior to this initiative, recruitment to clinical research in SLaM relied on the traditional system of researchers finding clinicians who were willing to identify and approach potential participants. Such nurse, medical and other clinicians may have little or no training in research and scant knowledge of any particular research programme; or they may themselves be researchers. This, together with clinical pressures and conflicting demands on time, result at best in the possibility of biased recruitment and, at worse, limit recruitment to research and obstruct patients from making decisions about participation.

The model is embedded within a search and database system (the Clinical Record Interactive Search [CRIS]), which reads and extracts information from SLaM's EHRs, removes identifiers and makes this available to researchers in de-identified format in standard analysis packages (eg, Stata, Excel). Data consist of all clinical records on SLaM patients (unless they have requested to opt out from the register) and are searchable as both structured and free text. There are currently over 250 000 cases on the database, which increases by approximately 20 000 per year. We have described the development and characteristics of CRIS elsewhere;[27] there is also published research based on analyses of CRIS data.[28] We present the features of CRIS that allow research participant recruitment.

### Reverse search
The case register was initially approved for use as a de-identified database whose data are searchable without consent by appropriately vetted researchers. The evaluation of this de-identification procedure demonstrated that CRIS effectively ensures patient anonymity at the same time as maximises data (free text and structured text) that are available for research. Indeed, our bespoke pattern matching de-identification algorithm (which is applied to all structured and free text in CRIS) was shown, when evaluated, to mask patient identifiers with 98.8% precision and 97.6% recall—outperforming a comparator machine learning algorithm. (We have published a full description of the algorithm and the evaluation data.)[29] The register's technical architecture included, additionally, the potential for reverse search: allowing the identification of patients who meet given characteristics (which can be defined using structured and free text), and thereby the possibility of using CRIS to identify and approach potential participants on the basis of prior consent by individual patients or, for children or adults lacking capacity, an appropriate proxy. In effect, such a mechanism allows for the creation within the case register of a database of people ('the recruitment database') who have provided prior consent to be contacted and whose full—but de-identified—clinical records will be available to a researcher (a 'recruiter') in order to search for inclusion/exclusion criteria for specific, ethically approved studies. The researcher, once they have identified eligible potential participants, can then be given identifiers to access the source EHRs and approach patients about participation in that particular study (figure 1).

We have developed robust procedural mechanisms to address legal and ethical requirements and to complement this process and technical design, which we briefly outline below.

### Acquiring and recording consent for contact
▶ The consent process for participation in the recruitment database is conducted by the patient's clinical team, most commonly by the patient's care

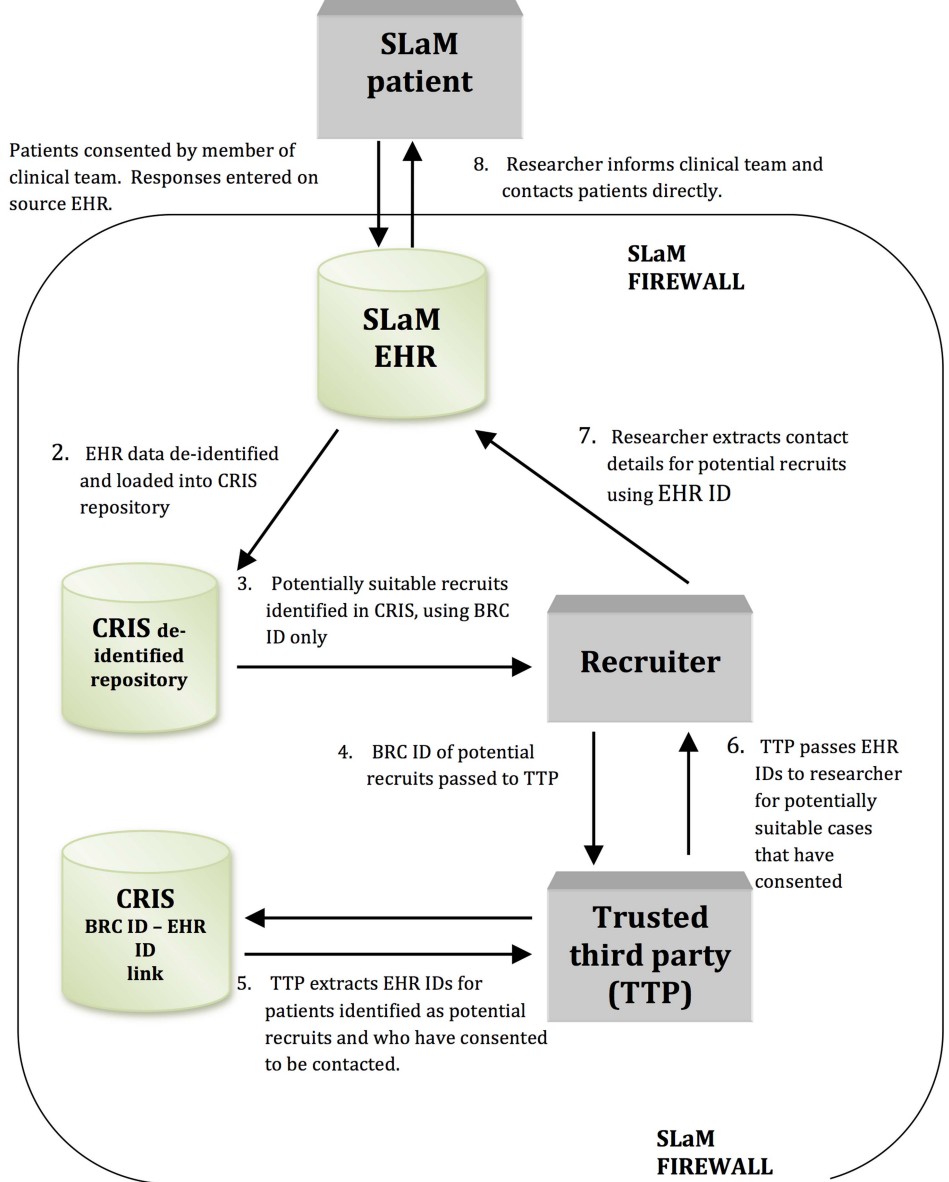

**Figure 1** The South London and Maudsley (SLaM) 'consent for contact' model.

coordinator (or, if not, by another member of the patient's care team). The process is carried out face to face. The SLaM C4C process is, additionally, being publicised across the Trust. If a SLaM patient expresses an interest in joining the C4C recruitment database by contacting a member of the SLaM C4C project team, that project worker (who is based in a clinical team and bound by the same information governance requirements as clinicians) is able to carry out the consent procedures. In addition, the patient's own care coordinator or equivalent clinician will be informed to check that there are no reasons why that patient should not be included, as well as encouraged to fill out the research participation form within the patient's electronic health record. In other words, there is always clinician involvement in the process of consenting a patient on to the C4C

recruitment database. The member of the clinical care team requesting consent receives training in the specifics of the SLaM C4C model and on how best to conduct the conversation clearly in a way that does not place undue pressure on the patient.

▶ Clinical staff are trained in research governance (particularly regarding consent and the assessment of mental capacity to provide consent) and the specific process of acquiring 'consent for contact'. The process, supported by an information sheet (see online supplementary appendices 1a–c), emphasises that patients are not being asked for consent to participate in any particular study—simply to being contacted in the future by researchers about potential participation in specified research projects based on information in their full SLaM EHR. The clinician explains, as required, more about what research is

and what it might entail; what tends to be recorded in an EHR; and reminds the patient that, if she consents, she might be asked to participate in research on different topics from those relating to her current treatment. A research study has been conducted to investigate how SLaM C4C consent conversations are carried out by clinicians and to ascertain which kinds of explanations of the recruitment database appear to make it more or less likely that patients will give their consent.[30]

▶ For patients who are children, or adults lacking capacity (where the lack of capacity is believed to be permanent), proxy assent is sought from a close friend or relative (with parental responsibility in the case of children; see online supplementary appendices 1b and 1c). The model is attentive to the mental health context for which it was designed: where mental capacity is thought to be temporarily absent, the clinical team does not ask for consent, but aims to approach the patient at a later time in accordance with the second principle of the Mental Capacity Act.[31]

▶ The approach made by a clinician requesting consent for contact is tailored to the clinical context—for example, the request may be delayed for a distressed patient admitted in a crisis.

▶ A research participation form has been created as an additional window in the source clinical records system (figures 2 and 3). This includes:
– Whether a C4C discussion has taken place with the patient (or a named proxy);
– Whether the patient (or proxy) gave consent or not;
– The name of the clinical staff member discussing C4C and the date discussed;
– A free text box to record any further information. Notably, patients are encouraged to identify any particular preferences regarding research opportunities —both those that they would be particularly interested in, and kinds of research, or research topics, that they would not want to be contacted about.

▶ For patients who have consented to being contacted, the research participation form is also used to record all subsequent contacts made by researchers, and all projects the patient is participating in, has participated in, and/or has declined to participate in. If a patient withdraws consent to be contacted then the technical ability to link to his or her EHR ID will be prevented.

▶ The model acknowledges that patients might not be aware, at the point of giving consent to participating

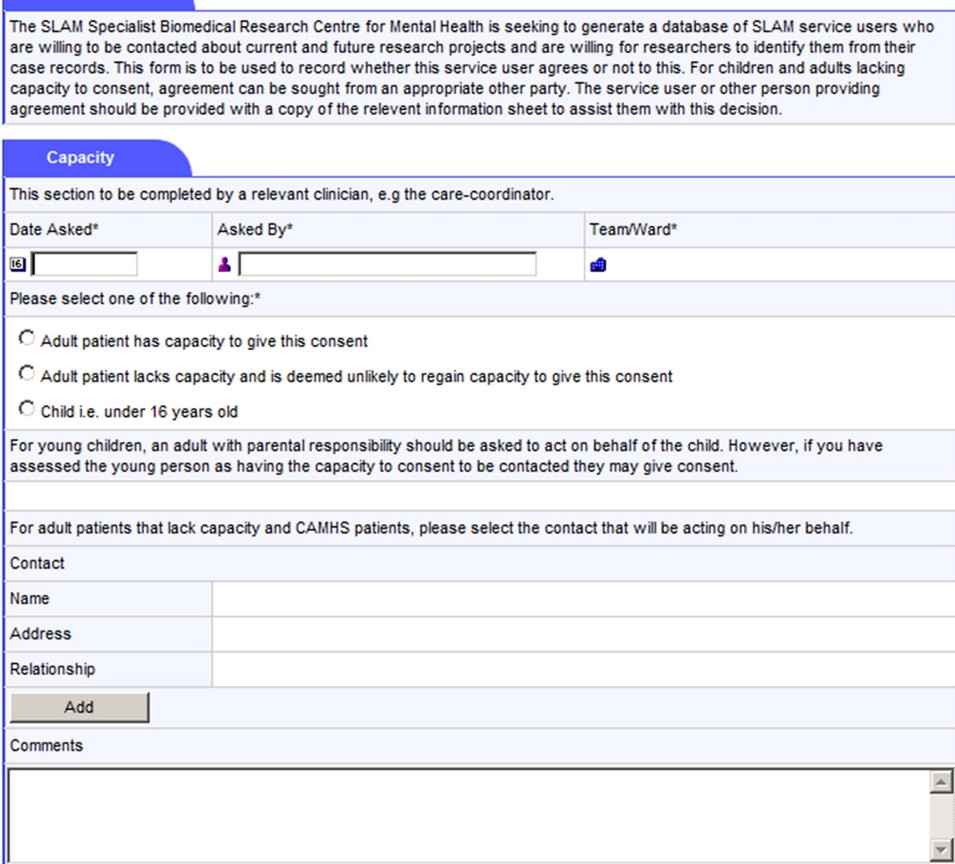

**Figure 2** Screen shot of the South London and Maudsley NHS Trust Consent for Contact model (SLaM C4C) Patient Participation Form in the original source Electronic Health Record (EHR; part I).

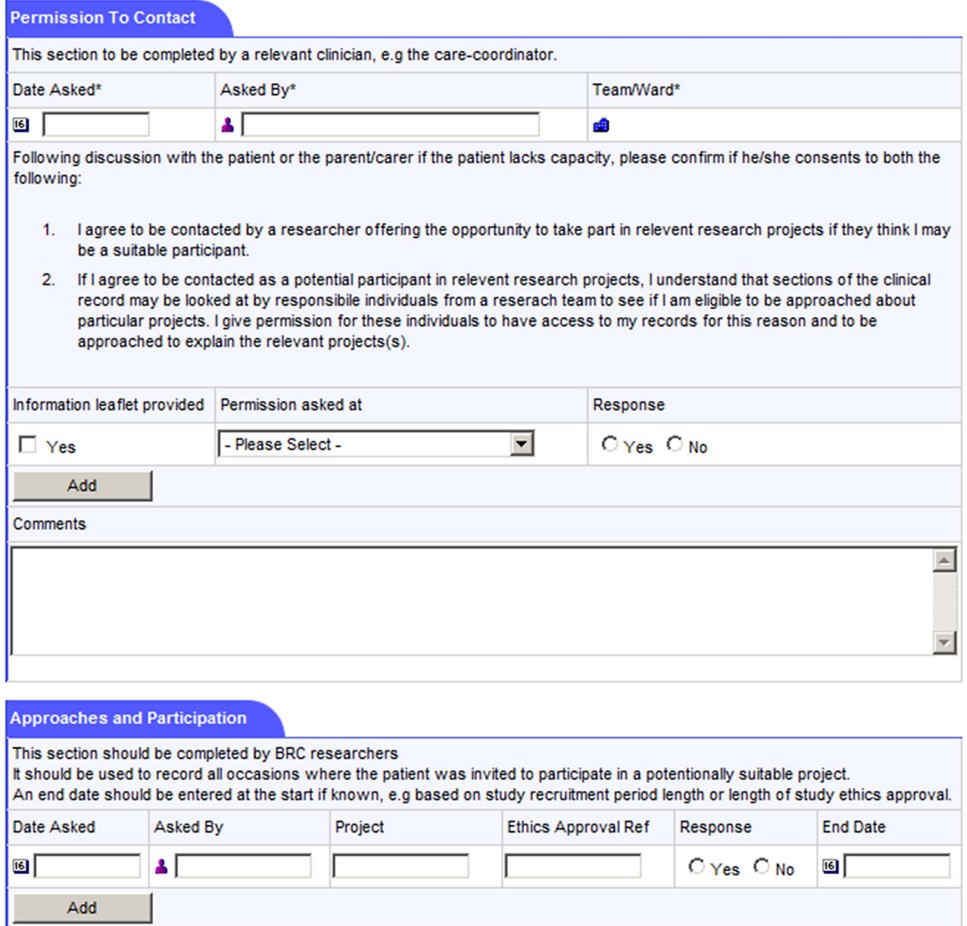

**Figure 3**  Screen shot of the South London and Maudsley NHS Trust Consent for Contact model (SLaM C4C) Patient Participation Form in the original source Electronic Health Record (EHR; part II).

in the recruitment database, what information will later be contained in their medical record. All consents are revisited with the participant at the point of discharge from all SLaM services or else are deemed to have expired.

### Identifying and contacting potential research participants

▶ In order to become a 'recruiter'—that is, have the ability to access CRIS for approaching potential research participants—the researcher must have a substantive or honorary contract with SLaM and be an employee of one of the organisations forming King's Health Partners, the Academic Health Sciences Centre (AHSC) of which SLaM is a member. Use of CRIS for all forms of research is logged and audited and misuse would result in disciplinary action.

▶ Recruiters use de-identified data on CRIS to identify potential recruits. In CRIS itself, records are identified by a locally generated pseudonym (the Biomedical Research Centre ID; BRC ID). This pseudonym is linked to the EHR ID number in the building of the CRIS data repository. Research users of CRIS are unable to access the link between BRC ID and EHR ID (figure 1), which maintains the integrity of CRIS as a de-identified database.

▶ An Oversight Committee (which is chaired by a mental health patient) manages all access to, and monitors all use of, CRIS, and reports to the Trust Caldicott Committee (which has responsibility for ensuring the protection of patient confidentiality throughout the Trust). The committee provides operational oversight and management of CRIS—including the provision of research governance for projects using CRIS; the monitoring and regular review of the effectiveness of the CRIS security model (including the de-identification processes); oversight of the administration of CRIS, including access control and maintenance, as well as the monitoring of audit logs; provision of advice on how to use CRIS; and responses to complaints related to CRIS (including from patients). There is no quorum required for individual meetings of the committee, but membership of the Committee must include patient/service user representation, a representative of the Trust's Caldicott Guardian, Trust research and development

(R&D) representation, and child and adolescent mental health services (CAMHS) representation. The Oversight Committee, its structure and its function are an integral component of the CRIS data resource as ethically approved.

▶ The Oversight Committee uses ethical, legal and scientific criteria to adjudicate such access and use. Applications to use CRIS to identify potential recruits are considered only for projects that have specific ethics and research governance approval. (In other words, projects using the SLaM C4C model must first acquire ethics and research governance approval and then approval from the Oversight Committee.) In the UK, acquiring this approval necessitates specifying how exactly patients will be recruited. This provides an additional layer of oversight, which helps to ensure that appropriate modes of approach are made to patients who have given 'consent for contact' when the topic of research is sensitive (eg, researchers might be required to talk to the treating clinician before contacting the patient).

▶ The recruiter submits the pseudonymised BRC ID of potential recruits they wish to contact to a trusted third party (TTP) appointed and monitored by the Oversight Committee.

▶ The TTP has project-specific access to the database linking the BRC IDs with the source EHR IDs. Technical specifications within the system ensure that CRIS cannot return the EHR ID of any patient who has not given consent for contact. Following reverse search, the TTP passes the EHR IDs of the potential recruits that have given consent back to the recruiter.

▶ The recruiter, through the SLaM EHR, is then able to identify and contact the patient to discuss participation in the project. Safeguards include a time limit on when the recruiter may contact patients about recruitment, and the requirement that recruiters inform each patient's care coordinator by email that they will be contacting that patient in a few days' time (eg, to allow the opportunity for the care coordinator to suggest that an approach at the current time might not be advisable).

▶ The Oversight Committee monitors the research participation forms to ascertain if approaches for research participation or actual research participation appear excessive. What would constitute 'excessive' is an ethical, sociological and scientific question: what might be excessive for certain individual patients, or groups of patients, might not be for others; participating in multiple surveys might be adjudicated differently from participation in several clinical trials in a short time frame. The Oversight Committee is formalising guidelines to adjudicate and respond to putatively 'excessive' research participation. Options for action include contacting the patient or clinical team about whether the patient continues to be willing to be approached about further research projects during the course of the current projects in which they are participating.

### Involving and engaging stakeholders

There is increasing acknowledgement of the importance of engaging stakeholders and the public in designing ethical and technically robust procedures to guide the use of EHRs.[8 10 11 32] Mental health service users and patients made significant contributions to the development of our model. The Patient and Carer Participation Theme within the SLaM BRC took a central role in developing the model; the Oversight Committee is chaired by a mental health patient; and consultation and engagement with service user/patient groups have taken place throughout, in line with our BRC's model of involving service users/patients at all stages of translational research.[33] An annual newsletter is sent to all persons in the recruitment database summarising key findings arising from the SLaM BRC, in addition to reminding people of contact details for the database (eg, if they wish later to withdraw). The SLaM C4C model has a dedicated website, and there are regular dissemination activities within the Trust to publicise the model. Details of how to contact the SLaM C4C team (who assist with requests to join the register, as well as provide information on how to withdraw from the register) are also disseminated online and via other media. Those who consented to join the register but have been discharged from the Trust are regularly sent a reminder that they are on the register as well as information on how to withdraw if they wish. As of July 2014, 30 patients have withdrawn their consent from being listed on the register (see box 2 for a summary of key features of the SLaM C4C model).

### RESULTS
#### Summary statistics from the implementation of SLaM C4C
SLaM C4C started to be implemented across the South London and Maudsley NHS Trust in May 2012, and the

---

**Box 2** Key features of the South London and Maudsley NHS Trust Consent for Contact (SLaM C4C) model

▶ Extensive patient involvement throughout model's development.
▶ The initial contact is made by a member of the patient's clinical care team.
▶ Researchers gain access to identifiable information only for patients who have given prior consent.
▶ All recruiting researchers have undergone extensive vetting procedures, and are employed on contracts that have the same level of duties and responsibilities (and same potential penalties re disciplinary action and dismissal) as those with substantive clinical contracts.
▶ Patients revisit their consent to be part of the recruitment database on discharge from the service.
▶ Regular (at least yearly) engagement with those consenting to be part of the recruitment database—about ongoing research studies, as well as reminders about their consenting to be part of the recruitment database and details on how to withdraw consent.

aim is to implement C4C across the entire Trust, which serves a total of approximately 35 000 active patients, split across seven clinical academic groups. All patients are gradually being approached and asked whether they will provide 'consent for contact'. There are no exclusion criteria. Implementation started in services where there was already support for and/or enthusiasm about C4C. The figures from 25 October 2013 show that a total of 2106 patients had been approached, of whom 1560 had given consent and 546 had not: a 74.1% consent rate. Table 1 presents descriptive statistics, from October 2013, of those approached to give consent for contact, in relation to gender, ethnicity and age.

## DISCUSSION

Initial data suggest that the SLaM C4C model is capable of accelerating the development of a culture of active research participation that is founded on the ethical and effective use of EHRs. The model has been successfully developed with significant patient/service user involvement, has received necessary approval and endorsement from all relevant governance bodies, and is currently resulting in almost three of every four patients approached agreeing to join the recruitment database. Public surveys and bioethical analyses commonly indicate that mental health is a sensitive domain in relation to EHRs, in light of concerns about potential stigma and discrimination.[32 34 35] The current consent figures in this secondary mental health services context are therefore encouraging vis-à-vis the model's transferability elsewhere.

Preliminary data from the implementation of SLaM C4C raise some interesting lines of investigation. That there are currently lower rates of consent from patients over 75 concurs with other research that indicates that older people are more likely to refuse participation in health research.[36] That there are currently higher rates of consent by men than women parallels other findings indicating that men are more likely to consent than women to a review of their medical records.[35] Variations in consent rates pose questions about possible selection bias in studies using this recruitment database. More research is needed to assess how many of those giving consent for contact subsequently consent to participate in particular research studies. Studies currently underway that are using the SLaM C4C model to recruit participants include: a longitudinal study to discover and validate biomarkers in Alzheimer's disease; an interventional randomised, double-blind exploratory study investigating the effects of an atypical antidepressant on cognition and BOLD fMRI signals in participants remitted from depression and controls; a study examining differences in cognitive appraisals of anomalous experiences and different facial emotions at a cognitive and neural level between individuals with psychotic symptoms with a need for care versus those without a need for care; and a study aiming prospectively to validate a set of questionnaires for the monitoring of treatment outcomes and side effects (including suicidality and self-harm) in general populations and in populations known to be at elevated risk of suicide.

The model that we have presented adheres to current best practice in recruiting patients,[24] most notably as regards ensuring that no identifiable patient information is available to researchers without the patient's consent. The model appears to be effective in its implementation—both in enabling the creation of a recruitment database, and in terms of acceptability to patients. We believe the model to be generalisable to other health services contexts that employ EHRs. Our intent when designing it was to guard against the erosion of trust in research—a key risk associated with the use of medical records without consent[34]—both through adhering to consent for contact principles (ie, ensuring that patients are explicitly asked for their consent before their medical records can be looked at for research purposes) and committing regularly to engage with members of the recruitment database about ongoing research and about their current willingness to be contacted about potential research possibilities. We designed the model to allow patients greater autonomy in decisions to take part in research, through lessening the gatekeeper role of clinicians: evidence from a prospective cohort study and a qualitative process evaluation indicates that such gatekeeper functions can impede equitable access to research.[13 37]

**Table 1** Descriptive data regarding patients approached for SLaM C4C

| | Total, (N) | Consenting, (N) | Consent rate (%) |
|---|---|---|---|
| Gender | | | |
| Male | 1078 | 844 | 78.3 |
| Female | 1028 | 716 | 69.6 |
| Self-assigned ethnicity (amalgamated) | | | |
| Caucasian | 1228 | 894 | 72.8 |
| Caribbean, African or any other black background | 496 | 360 | 72.6 |
| Not stated | 205 | 168 | 82.0 |
| Other | 100 | 78 | 78.0 |
| Indian, Pakistani, Bangladeshi or any other Asian background | 77 | 60 | 77.9 |
| Age* | | | |
| 0–19 | 841 | 667 | 79.3 |
| 20–29 | 182 | 165 | 90.7 |
| 30–44 | 232 | 213 | 91.8 |
| 45–74 | 370 | 283 | 76.5 |
| 75+ | 481 | 232 | 48.2 |

Data captured on 25 October 2013.
*Age distribution is not representative of the Trust as a whole, since implementation of SLaM C4C to date has significantly focused on Mental Health of Older Adults Services, and Child and Adolescent Mental Health Services.

## A new culture of research participation?

The use, linkage and further development of large EHR data sets are likely to transform relations between researchers, clinicians, patients and their data. The implementation of SLaM C4C, across a large mental health provider, generates, we believe, effectively a field site or laboratory in which to study these potential transformations. The Oversight Committee regularly captures descriptive data that give broad indications of how implementation of SLaM C4C is proceeding (both as regards patients approached, and as regards the number and type of research studies applying to use SLaM C4C). In addition, SLaM BRC is planning additional research and evaluation studies (see box 3 for indicative research studies that address important questions associated with the use of EHRs for health research).[32 35]

Our BRC's ongoing collection of survey and audit data, as well as planned research studies, comprise a powerful means through which to interrogate and analyse the sociological, ethical, technical and governance-related ramifications of large-scale EHR implementation of consent for contact, in which clinicians no longer provide the primary conduit for patient participation in research.

---

**Box 3** Research and evaluation questions raised by the South London and Maudsley NHS Trust Consent for Contact (SLaM C4C) model

► *Conveying consent for contact to patients.* One current study is using focus groups and reiterative methods to ask: What kinds of explanations of consent for contact are most clearly comprehensible to patients?

► *Patients' experience of consent for contact.* How do patients interpret the process of giving (or refusing) generic consent to access medical records? Does the SLaM C4C model affect their sense of autonomy vis-à-vis decisions about participating in health research? How often do those who give consent for contact go on to participate in research?

► *Researchers' experience of consent for contact.* Does it facilitate recruitment (in terms of ease of recruiting, time to reach recruitment targets)? Does the possibility of being able to use free text in CRIS to identify and cluster patients change how studies are designed?

► *Clinicians' experience of consent for contact (both those who make the approach for 'consent for contact' and others).* How, practically, do clinicians carry out and record consent for contact discussions with their patients? Does consent for contact alter relationships between clinicians and patients?

► *Differential rates of those consenting.* How and why do rates of consent vary across demographic groups and types of service? Does this produce selection bias in studies recruiting from the database?

► *Consent for contact as a potential mechanism to help facilitate public engagement vis-à-vis participation in health research and broader use of, and trust in, Electronic Health Records (EHRs).*

---

**Author affiliations**
[1]Centre for Medical Humanities and Department of Geography, Durham University, Durham, UK
[2]King's College London, SLaM Biomedical Research Centre Nucleus, Institute of Psychiatry, Psychology and Neuroscience, London, UK
[3]Oxford Academic Health Science Network (AHSN), Oxford, UK
[4]Department of Psychological Medicine, King's College London, Institute of Psychiatry, Psychology and Neuroscience, London, UK
[5]South London and Maudsley NHS Foundation Trust, London, UK
[6]Department of Psychology, King's College London, Institute of Psychiatry, Psychology and Neuroscience, London, UK
[7]Department of Psychiatry, University of Oxford, Warneford Hospital, Oxford, UK
[8]King's College London, Institute of Psychiatry, Psychology and Neuroscience, London, UK

**Acknowledgements** The authors thank the NIGB Ethics and Confidentiality Committee for its helpful comments on and input into the final design of the model for recruitment. They also thank the CRIS Oversight Committee and the CRIS Security and Confidentiality Procedures Working Group. Andrea Fernandes (CRIS Administrator), Jenny Liebscher (Research & Development [R&D] Governance and Delivery Manager for the SLaM and Institute of Psychiatry, Psychology and Neuroscience R&D Office), Sheri Oduola (Consent for Contact (C4C) Project Manager) and Megan Pritchard (CRIS Training and Development Lead) provided important details on how the SLaM C4C model is being implemented in practice.

**Contributors** The idea for the manuscript was conceived by FC in conversation with RS and SL. FC drafted the manuscript, with substantial text contributions from MB, SL and RS. All authors were involved in critical revision of the manuscript before submission, and approved the final manuscript. All authors developed and/or refined the consent for contact model described in the paper. FC is the guarantor. FC is Chair of the CRIS Oversight Committee and was Chair of the Security and Confidentiality Procedures Working Group that developed the security model for CRIS; MB project managed the development of CRIS; MD directed information strategy within SLaM during the development of CRIS and SLaM C4C; MH developed the protocol for patients lacking mental capacity; MS is Head of Information Governance and the Caldicott representative at SLaM; TW is Patient and Carer Participation Theme Lead for the Biomedical Research Centre for Mental Health; SL was director of the Biomedical Research Centre for Mental Health during the development of CRIS and SLaM C4C; RS led the academic development of CRIS. RS, SL and MH secured funding for the work.

**Funding** The authors acknowledge financial support from the National Institute for Health Research (NIHR) Specialist Biomedical Research Centre for Mental Health award to the South London and Maudsley NHS Foundation Trust and the Institute of Psychiatry, Psychology and Neuroscience, King's College London. The development of the SLAM BRC Case Register was funded by a Capital Award from the UK NIHR and is further supported through the BRC Nucleus funded jointly by the Guy's and St Thomas' Trustees and South London and Maudsley Special Trustees. FC, MB, MH, TW, SL and RS—as well as CRIS itself—are or have been part-funded by the National Institute for Health Research (NIHR) Biomedical Research Centre and Dementia Biomedical Research Unit at South London and Maudsley NHS Foundation Trust and King's College London for this research. SL and TW acknowledge the NIHR for their Senior Investigator Awards. FC receives salary support from The Wellcome Trust (WT103817MA). The funders had no role in study design, data collection and analysis, decision to publish, or preparation of the manuscript.

**Competing interests** SL is co-ordinator of the European Medical Information Framework (http://www.emif.eu), a public private partnership with funding from EU and pharmaceutical companies that seeks to make use of both research and routine clinical data for research.

**Ethics approval** The SLaM C4C model was reviewed by the National Information Governance Board for Health and Social Care (NIGB) Ethics and Confidentiality committee (reference ECC 2–08/2010); the committee considered the model 'an elegant solution to the issue of participant recruitment' and stated that it 'strongly endorse[s] this approach to participant

recruitment'. The model was approved by a Research Ethics Committee (REC) (South East London 4 REC, Ref 10/H0807/88), as well as by the SLaM Caldicott Guardian and the SLaM Trust Executive. The Clinical Record Interactive Search (CRIS), around which the model is built, was approved for use by a REC (Oxford C REC, Ref 08/H0606/71+5); its security model (which comprises both technical and procedural elements) was approved by the SLaM Trust Caldicott Guardian, and signed off by the Trust Executive.

**Provenance and peer review** Not commissioned; externally peer reviewed.

**Data sharing statement** No additional data are available.

**Open Access** This is an Open Access article distributed in accordance with the terms of the Creative Commons Attribution (CC BY 4.0) license, which permits others to distribute, remix, adapt and build upon this work, for commercial use, provided the original work is properly cited. See: http://creativecommons.org/licenses/by/4.0/

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
