## [Reviewer comments · BMJ Open]

ARTICLE DETAILS

TITLE (PROVISIONAL)	Developing a new model for patient recruitment in mental health services: a cohort study using electronic health records
AUTHORS	Callard, Felicity; Broadbent, Matthew; Denis, Mike; Hotopf, Matthew; Soncul, Murat; Wykes, Til; Lovestone, Simon; Stewart, Robert

VERSION 1 - REVIEW

REVIEWER	Prof Jacqueline Atkinson University of Glasgow Scotland UK
REVIEW RETURNED	05-Jun-2014

GENERAL COMMENTS	this is a very minor quibble. the SLaM C4C is described very well. what i am slightly unclear about is the part of the study which asks patients if they will consent to allow their records to be used. How ere patients approached - in person, if so by whom? by letter? It only needs 1-2 lines to clarify this bit of the methodology
--

REVIEWER	Donald Willison University of Toronto, Institute for Health Policy, Management and Evaluation, Dalla Lana School of Public Health, Canada.
REVIEW RETURNED	16-Jun-2014

GENERAL COMMENTS	The manuscript has made substantial improvements over the previous version, in response to reviewers' comments. This reviewer has a few remaining comments to this revised version. 1. On page7, lines 37 and 38 indicate that this may be the first successful implementation of consent for contact. While this particular variant using de-identified electronic health records may be a first, the authors should be aware of recent papers describing a similar project in British Columbia that began in 2007. Below are the references. In addition, this reviewer is aware of other institutions that have developed a similar process but which have not published their findings. Nonetheless, the detailed description of this project and the issues addressed merit publication of this relatively unique work. The British Columbia references are: (1) Cheah S, O'Donoghue S, Daudt H, Dee S, LeBlanc J, Braun L, et al. Permission to Contact (PTC)-A Strategy to Enhance Patient Engagement in Translational Research. Biopreservation and biobanking 2013;11(4):245-252. (2) LeBlanc J, Dee S, Braun L, Daudt H, Cheah S, Watson PH. Impact of a Permission to Contact (PTC) Platform on Biobank Enrollment and Efficiency. Biopreservation and biobanking 2013;11(3):144-148.
---

	2. In the beginning of the manuscript, there appears to be some blurring between the issues of permission to review the patient’s medical record and permission to contact the patient about future research. Once the Clinical Record Interactive Search (CRIS) is explained, it seems as if the issue of consent to review the record is obviated by the anonymization of the record in the database. However, when one reviews the patient communication material in the appendix, it appears as if this blurring has a history. In the patient communication document and the corresponding document for people responsible for other adults, the first and fifth paragraphs both indicate that permission is being sought to review the record – albeit with a view to contacting them in future. Yet, the consent form itself (Figure 5), is clearly a consent to be contacted in future. It is important that the authors keep clear this distinction. Would the authors please explain the discrepancy between the communication documents and the consent form? 3. On page 9, line 9 and following, the CRIS database that contains information from the patient record is mentioned and referenced for the reader to learn more about things like how the record is de-identified. On reviewing reference 24 (by R.Stewart et al), there is a very high-level description of how the quantitative and directly identifying data are de-identified, but the authors only acknowledge that anonymization of the free-text data is a greater challenge. Would the authors please clarify what measures are currently taken to de-identify the free-text data in the CRIS database? 4. P. 11, lines 33-34: Please clarify how, and at what points, the process of “revisiting” the consent to be contacted occur. Figure 3 indicates at least an annual engagement regarding continued participation. Is this provided with the annual newsletter? What is the process for this engagement and for withdrawal from participation? 5. P. 12. re: Method of first contact: In their response to my initial reviewer’s comments about details around first contact, the authors provided a reasonably comprehensive response. However, the description in the text refers only to the contacting of the patient’s care coordinator. A more fulsome discussion is warranted. 6. P. 12, lines 10-20: Please describe in greater depth the Oversight Committee. In particular: (a) Is the remit of the Oversight Committee solely over access to CRIS? (b) Do they provide formal approval separate from REC approval? If so, is their approval required as a condition of REC approval? (c) The description of membership is somewhat vague. Are there certain members that must be present for a quorum? 7. Minor: Language is not entirely consistent around “patient” vs. “service user” – e.g. in Figure 3. This is a paper that will be of high value to a niche audience interested in facilitating the recruitment process in an ethically acceptable fashion.
--	---

REVIEWER	Clare Relton University of Sheffield
REVIEW RETURNED	21-Jun-2014

GENERAL COMMENTS	Thank you for asking me to review this interesting article describing a new model of identifying and recruiting patients to research using 'consent for contact' and electronic health records. I enjoyed reading this. The article describes the implementation of the innovative SLaM C4C model in a large mental health setting. The potential of this approach to better engage with potential research participants is as important as the efficiency of this method of identifying and approaching potential research participants. A very brief description of the types of studies that have recruited through the SLaM c4c model would be of interest to the reader. A sentence in the introduction on the prevalence of the 'consent for contact' model in other research facilities (such as Biobank) might be illuminating for the reader. Page 3 line 14 - remove 'to' replace with 'a' line 14 - suggest a model for identification and recruitment line 17 - this might be better described as 'reduce the burdensome (or time consuming and cumbersome) gatekeeper role of clinicians line 24 - approached by whom? Page 6 line 26 - remove 'urgent scientific' Line 52 - 57 - not sure this information is required Figure 1 - I suggest that each bullet point is framed as 'How does.....' Page 8 line 9 - which REC?? line 26 - which rigorous approvals are you referring to here? Page 10 line 24 - 'reminds the patient Page 16 line 8 - can you make explicit what you mean by 'consent for contact' principles lines 16 - 19 - this phrase very succinctly summarises the rationale for the SLaM model - can you use this in your introduction (the introduction section at the moment could be made more succinct perhaps?)
--

VERSION 1 – AUTHOR RESPONSE

Reviewer: 1

Reviewer Name Prof Jacqueline Atkinson

Institution and Country University of Glasgow

Scotland UK

Please state any competing interests or state 'None declared': none declared

this is a very minor quibble.

the SLaM C4C is described very well.

what i am slightly unclear about is the part of the study which asks patients if they will consent to allow their records to be used.

How ere patients approached - in person, if so by whom? by letter?

It only needs 1-2 lines to clarify this bit of the methodology

We have added significantly more detail on pp.10–11 to clarify that the process is conducted face to face, and always involves a member of the clinical care team in some way:

- ‘The consent process for participation in the recruitment database is conducted by the patient’s clinical team, most commonly by the patient’s care coordinator (or, if not, by another member of the patient’s care team). The process is carried out face to face. The SLaM C4C process is, additionally, being publicized across the Trust. If a SLaM patient expresses an interest in joining the C4C register by contacting a member of the SLaM C4C project team, that project worker (who is based in a clinical team and bound by the same information governance requirements as clinicians) is able to carry out the consent procedures. In addition, the patient’s own care coordinator or equivalent clinician will be informed to check that there are no reasons for why that patient should not be included, as well as encouraged to fill out the research participation form within the patient’s electronic health record. In other words, there is always clinician involvement in the process of consenting a patient on to the C4C register. The member of the clinical care team requesting consent receives training in the specifics of the SLaM C4C model and on how best to conduct the conversation clearly in a way that does not place undue pressure on the patient.’

Reviewer: 2

Reviewer Name Donald Willison

Institution and Country University of Toronto, Institute for Health Policy, Management and Evaluation, Dalla Lana School of Public Health, Canada.

Please state any competing interests or state ‘None declared’: None declared.

The manuscript has made substantial improvements over the previous version, in response to reviewers’ comments. This reviewer has a few remaining comments to this revised version.

1. On page7, lines 37 and 38 indicate that this may be the first successful implementation of consent for contact. While this particular variant using de-identified electronic health records may be a first, the authors should be aware of recent papers describing a similar project in British Columbia that began in 2007. Below are the references. In addition, this reviewer is aware of other institutions that have developed a similar process but which have not published their findings. Nonetheless, the detailed description of this project and the issues addressed merit publication of this relatively unique work. The British Columbia references are: (1) Cheah S, O’Donoghue S, Daudt H, Dee S, LeBlanc J, Braun L, et al. Permission to Contact (PTC)-A Strategy to Enhance Patient Engagement in Translational Research. *Biopreservation and biobanking* 2013;11(4):245-252. (2) LeBlanc J, Dee S, Braun L, Daudt H, Cheah S, Watson PH. Impact of a Permission to Contact (PTC) Platform on Biobank Enrollment and Efficiency. *Biopreservation and biobanking* 2013;11(3):144-148.

We thank the reviewer for this helpful information. We have inserted both the British Columbia project references, and added the following sentence on p.6:

‘Notably, ‘consent for contact’ processes are being developed in a range of health research facilities. For example, a project in British Columbia, Canada has, since 2007, set up ‘Permission to Contact’ platforms in different outpatient health clinics (cancer, cardiac and maternal health), which have proved effective in enhancing enrollment into translational research projects.[15 16]’

2. In the beginning of the manuscript, there appears to be some blurring between the issues of permission to review the patient’s medical record and permission to contact the patient about future research. Once the Clinical Record Interactive Search (CRIS) is explained, it seems as if the issue of consent to review the record is obviated by the anonymization of the record in the database. However, when one reviews the patient communication material in the appendix, it appears as if this blurring has a history. In the patient communication document and the corresponding document for

people responsible for other adults, the first and fifth paragraphs both indicate that permission is being sought to review the record – albeit with a view to contacting them in future. Yet, the consent form itself (Figure 5), is clearly a consent to be contacted in future. It is important that the authors keep clear this distinction. Would the authors please explain the discrepancy between the communication documents and the consent form?

The SLaM C4C model involves both consent for researchers to review the entirety of the medical record and consent vis-à-vis being potentially contacted in the future by researchers about particular research studies for which the patient might be appropriate. The researchers initially search CRIS as an anonymised database and are able to identify potential participants who fulfill their criteria. They then hand over the list of BRC IDs to the safe haven – and will only be given back the NHS IDs of those patients who have given SLaM C4C consent. We believe that both the patient communication document and the research participation form (which is a computerized form held within the patient's electronic health record) make clear that permission is being given for two things to happen if the patient gives his/her consent. In the 'Permission to Contact' section in Figure 4 (p.26), the patient is (i) agreeing to being contacted AND (ii) giving permission for his/her medical record to be looked at by a researcher; in the communication materials (pp. 28ff.), it is made clear that permission is being asked to review medical records, and that if this permission is given and the person joins the list, s/he is agreeing to researchers potentially contacting him/her (see p. 29 'What will happen if I agree to be on the list').

3. On page 9, line 9 and following, the CRIS database that contains information from the patient record is mentioned and referenced for the reader to learn more about things like how the record is de-identified. On reviewing reference 24 (by R.Stewart et al), there is a very high-level description of how the quantitative and directly identifying data are de-identified, but the authors only acknowledge that anonymization of the free-text data is a greater challenge. Would the authors please clarify what measures are currently taken to de-identify the free-text data in the CRIS database?

We have added additional sentences on p. 9 (in the 'Reverse Search' section) reporting data from our publication on de-identification of CRIS (specifically addressing the issue of free text), namely:

'Indeed, our bespoke pattern matching de-identification algorithm (which is applied to all structured and free text in CRIS) was shown, when evaluated, to mask patient identifiers with 98.8% precision and 97.6% recall – outperforming a comparator machine learning algorithm. (We have published a full description of the algorithm and the evaluation data.)[26]'

4. P. 11, lines 33-34: Please clarify how, and at what points, the process of "revisiting" the consent to be contacted occur. Figure 3 indicates at least an annual engagement regarding continued participation. Is this provided with the annual newsletter? What is the process for this engagement and for withdrawal from participation?

We have added the following to p.15:

'The SLaM C4C model has a dedicated website, and there are regular dissemination activities within the Trust to publicize the model. Details of how to contact the SLaM C4C team (who would assist with requests to join the register, or provide information on how withdraw from the register) are also disseminated online and via other media. Those who consented to join the register but have been discharged from the Trust are regularly sent a reminder that they are on the register as well as information on how to withdraw if they wish. As of July 2014, 30 patients have withdrawn their consent from being listed on the register.'

5. P. 12. re: Method of first contact: In their response to my initial reviewer's comments about details

around first contact, the authors provided a reasonably comprehensive response. However, the description in the text refers only to the contacting of the patient's care coordinator. A more fulsome discussion is warranted.

We have added the following on pp.10–11 (see underlined passages below):

'Acquiring and recording consent for contact

- The consent process for participation in the recruitment database is conducted by the patient's clinical team, most commonly by the patient's care coordinator (or, if not, by another member of the patient's care team). The process is carried out face to face. The SLaM C4C process is, additionally, being publicized across the Trust. If a SLaM patient expresses an interest in joining the C4C register by contacting a member of the SLaM C4C project team, that project worker (who is based in a clinical team and bound by the same information governance requirements as clinicians) is able to carry out the consent procedures. In addition, the patient's own care coordinator or equivalent clinician will be informed to check that there are no reasons for why that patient should not be included, as well as encouraged to fill out the research participation form within the patient's electronic health record. In other words, there is always clinician involvement in the process of consenting a patient on to the C4C register. The member of the clinical care team requesting consent receives training in the specifics of the SLaM C4C model and on how best to conduct the conversation clearly in a way that does not place undue pressure on the patient.
- Clinical staff are trained in research governance (particularly regarding consent and the assessment of mental capacity to provide consent) and the specific process of acquiring 'consent to contact'. The process, supported by an information sheet (see Appendices 1a–c), emphasises that patients are not being asked for consent to participate in any particular study – simply to being contacted in the future by researchers about potential participation in specified research projects based on information in their full SLaM EHR. The clinician explains, as required, more about what research is and what it might entail; what tends to be recorded in an EHR; and reminds the patient that, if she consents, she might be asked to participate in research on different topics from those relating to her current treatment. A research study has been conducted to investigate how SLaM C4C consent conversations are carried out by clinician and to ascertain which kinds of explanations of the recruitment database appear to make it more or less likely that patients will give their consent.[27]

6. P. 12, lines 10-20: Please describe in greater depth the Oversight Committee. In particular: (a) Is the remit of the Oversight Committee solely over access to CRIS? (b) Do they provide formal approval separate from REC approval? If so, is their approval required as a condition of REC approval? (c) The description of membership is somewhat vague. Are there certain members that must be present for a quorum?

On p. 13, we have inserted the following sentences:

- 'An Oversight Committee (which is chaired by a mental health patient/service user) manages all access to, and monitors all use of, CRIS, and reports to the Trust Caldicott Committee (which has responsibility for ensuring the protection of patient confidentiality throughout the Trust). The committee provides operational oversight and management of CRIS – including the provision of research governance for projects using CRIS; the monitoring and regular review of the effectiveness of the CRIS security model (including the de-identification processes); oversight of the administration of CRIS, including access control and maintenance, as well as the monitoring of audit logs; provision of advice on how to use CRIS; and responses to complaints related to CRIS (including from patients). There is no quorum required for individual meetings of the committee, but membership of the Committee must include patient/service user representation, a representative of the Trust's Caldicott Guardian, Trust research and development (R&D) representation and child and adolescent mental health services (CAMHS) representation. The Oversight Committee, its structure and its function are

an integral component of the CRIS data resource as ethically approved.'

We have also added, on p.13, the clarifying sentence:

• '(In other words, projects using the SLAM C4C model must first acquire ethics and research governance approval and then approval from the Oversight Committee.)'

7. Minor: Language is not entirely consistent around "patient" vs. "service user" – e.g. in Figure 3.

We have systemized use (almost entirely replacing 'service user' with patient, except on a few occasions on which service user is used alongside patient [in deference to the self-descriptors of the groups referred to]).

This is a paper that will be of high value to a niche audience interested in facilitating the recruitment process in an ethically acceptable fashion.

We are grateful for the reviewer's assessment of the value of the paper to a particular audience.

Reviewer: 3

Reviewer Name Clare Relton

Institution and Country University of Sheffield

Please state any competing interests or state 'None declared': None

Thank you for asking me to review this interesting article describing a new model of identifying and recruiting patients to research using 'consent for contact' and electronic health records.

I enjoyed reading this.

The article describes the implementation of the innovative SLAM C4C model in a large mental health setting.

The potential of this approach to better engage with potential research participants is as important as the efficiency of this method of identifying and approaching potential research participants.

A very brief description of the types of studies that have recruited through the SLAM c4c model would be of interest to the reader.

We have inserted the following few sentences to give an indication of the kinds of studies that are currently recruiting using the SLAM C4C model on pp. 17–18:

'Studies currently underway that are using the SLAM C4C model to recruit participants include: a longitudinal study to discover and validate biomarkers in Alzheimer's Disease; an interventional randomised, double blind exploratory study investigating the effects of an atypical antidepressant) on cognition and BOLD fMRI signals in subjects remitted from depression and controls; a study examining differences in cognitive appraisals of anomalous experiences and different facial emotions at a cognitive and neural level between individuals with psychotic symptoms with a need for care versus those without a need for care; and a study aiming prospectively to validate a set of questionnaires for the monitoring of treatment outcomes and side-effects (including suicidality and self-harm) in general populations and in populations known to be at elevated risk of suicide.'

A sentence in the introduction on the prevalence of the 'consent for contact' model in other research facilities (such as Biobank) might be illuminating for the reader.

We have inserted the following sentence into the introduction (pp. 6–7):

'Notably, 'consent for contact' processes are used in a range of health research facilities: for example, the UK Biobank project requires explicit consent in order both to access the medical records of those

joining the project, as well as potentially to re-contact these participants in the future.[15]

Page 3 line 14 - remove 'to' replace with 'a'
Corrected.

line 14 - suggest a model for identification and recruitment
Corrected – we have used this phrasing suggestion.

line 17 - this might be better described as 'reduce the burdensome (or time consuming and cumbersome) gatekeeper role of clinicians'
Corrected – we have used the suggested phrasing.

line 24 - approached by whom?
We have lengthened the phrase: “approached by a member of their clinical care team using the ‘consent for contact’ model”.

Page 6

line 26 - remove 'urgent scientific'
Changed as suggested.

Line 52 - 57 - not sure this information is required

We would prefer to retain this sentence, since we believe that our SLaM C4C model marks a distinct shift from the preceding model (where it is the clinician who is alerted and then invited to contact the patient) – and hence that this contextualizing information is important.

Figure 1 - I suggest that each bullet point is framed as 'How does.....'

We have changed the first 2 bullets to ‘How does’ (we think that the other bullet points do not to require a ‘How does’ formulation).

Page 8

line 9 - which REC??

We have provided specific details of the REC that approved the SLaM C4C model, and approved the use of CRIS as an anonymised database, namely:

The SLaM C4C model was approved by: South East London 4 REC, Ref 10/H0807/88.

The use of CRIS as an anonymised database was approved by: Oxford C REC, Ref 08/H0606/71+5.

line 26 - which rigorous approvals are you referring to here?

On p. 8 we have inserted:

‘(These researchers have undergone rigorous approval procedures, which include being bound by the duties associated with a contract with King’s Health Partners [Academic Health Sciences Center], which in turn involves various Human Resources checks, including a criminal record check in accordance with national Department of Health standards.)’

Page 10

line 24 - 'reminds the patient'
Corrected.

Page 16

line 8 - can you make explicit what you mean by 'consent for contact' principles

We have added the following contextualizing phrase to clarify that we mean to juxtapose our approach (which uses 'Consent for Contact') with an approach that does not ask patients' consent for researchers to look through their records:

'both through adhering to consent for contact principles (i.e. ensuring that patients are explicitly asked for their consent before their medical records can be looked at for research purposes) and committing regularly ...'

lines 16 - 19 - this phrase very succinctly summarises the rationale for the SLAM model - can you use this in your introduction (the introduction section at the moment could be made more succinct perhaps?)

We have added the following sentence on p.5 (in the introductory section, at the end of the paragraph):

'In short, it is time to allow patients greater autonomy in decisions to take part in research by reducing the gatekeeper role of clinicians – not least because such gatekeeper functions can exacerbate inequities in who is able to participate in health research.'

VERSION 2 – REVIEW

REVIEWER	Jacqueline Atkinson University of Glasgow UK
REVIEW RETURNED	19-Aug-2014

GENERAL COMMENTS	minor point - I would avoid saying '-in other words ...' - you do this a couple of times say once, clearly - otherwise gets repetitive
--

REVIEWER	Donald Willison University of Toronto, Institute for Health Policy, Management and Evaluation, Dalla Lana School of Public Health, Canada.
REVIEW RETURNED	20-Aug-2014

GENERAL COMMENTS	I am satisfied with revisions to the manuscript and look forward to its publication.
--